# Effects of the Off-Label Drug Prescription in the Paediatric Population in Spain from the Adoption of the Latest European Regulation: A Pre-Post Study

**DOI:** 10.3390/pharmaceutics13040588

**Published:** 2021-04-20

**Authors:** Irene Lizano-Díez, Itziar Aldalur-Uranga, Carlos Figueiredo-Escribá, Cecilia F. Lastra, Eduardo L. Mariño, Pilar Modamio

**Affiliations:** Clinical Pharmacy and Pharmaceutical Care Unit, Department of Pharmacy and Pharmaceutical Technology and Physical Chemistry, Faculty of Pharmacy and Food Sciences, University of Barcelona, Av. Joan XXIII 27-31, 08028 Barcelona, Spain; ilizano@ub.edu (I.L.-D.); ialdalur7@ub.edu (I.A.-U.); cdefigueiredoescriba@ub.edu (C.F.-E.); ceciliafernandez@ub.edu (C.F.L.); emarino@ub.edu (E.L.M.)

**Keywords:** off-label prescription, paediatrics, paediatric regulation, pharmaceutical products, prescription drugs, primary health care

## Abstract

The year 2021 marks the 15th anniversary of the Paediatric Regulation (1901/2006/EC) in Europe. The main aim of the study was to conduct a pre-post comparison on the annual off-label prescription rates in the under-18 population in Spain and assess the potential influence of the Paediatric Regulation adoption. An observational study in the paediatric population was performed. Four cross-sectional annual periods, one before and the three latest periods after the adoption of the Regulation, were compared. Prescriptions in the primary health care setting were sorted by age group and drug and off-label status were determined. The number of off-label prescriptions issued by paediatricians was over two million per year. Prior to the adoption of the Paediatric Regulation, the off-label prescription rate was estimated at 7% of total prescriptions. Although the increase in the off-label rate over the study periods was mild, it was statistically significant (OR: 1.045; 95% CI: 1.043–1.046; *p* < 0.05). One of the most vulnerable population groups was neonates and infants up to 1 year, in which the off-label prescription rates showed the highest increase during the post follow-up period, which was statistically significant (OR: 4.270; 95% CI: 4.253–4.287; *p* < 0.05). The findings can help raise awareness and advocate for the development and authorization of medicines for children in the primary health care setting.

## 1. Introduction

The year 2021 marks the 15th anniversary of the Paediatric Regulation (1901/2006/EC) in Europe [1]. The aim of this regulation was to ensure that appropriate studies in children would be carried out in order to obtain the data necessary to support authorizing medicines in the paediatric population. This regulation raised the need to establish similar standards for paediatric medicines as for those used in adults, in terms of undergoing studies to ensure that medicines are safe, effective and high quality to be used in the paediatric population. Problems that commonly arise in current clinical practice, such as those related to the lack of medicines suitably adapted to the needs of the paediatric population (i.e., incomplete or insufficient dosage information that could be associated with an increased risk of adverse drug events, under-dosage, etc.) should be avoided or, at least, minimized. The European Medicines Agency (EMA) established procedures for paediatric investigation plans (PIPs), the documents that should serve as a framework for the development and authorization of medicinal products for children. Included in these PIPs are the requirements for adding a new indication, pharmaceutical form, or route of administration for an already authorized medicine covered by intellectual property rights [2].

According to the European Federation of Pharmaceutical Industries and Associations (EFPIA), medicine development in clinical trials takes an average of 12 years until approval for use in patients [3]. Under this scenario, the outcomes of the PIPs in innovations could have begun to be noted from year 2019. The latest update released by the European Commission (August 2020) shows that the Paediatric Regulation (1901/2006/EC) has resulted in an increase of almost 50% in clinical trials including children and in over 1000 PIPs agreed. The number of paediatric products authorized also increased after the adoption of the Regulation. By 2016, 101 paediatric medicines and 99 new paediatric indications had been centrally authorized. In the same period, 10 new paediatric medicines received a national authorization and 57 new paediatric indications were added to nationally authorized products [4].

In February 2017, the European Union (EU) published a study report on the off-label use of medicinal products in the ten years following the adoption of the Paediatric Regulation, which estimated that the regulation did not seem to have led to a lower prevalence of off-label use in the EU, although the number of medicines developed for children had increased during the ten-year period. The areas with the most progress were those in which there was an overlap between the needs of paediatric patients and those of adults, with far fewer achievements in rare diseases or diseases that mainly occur in children [5]. In 2018, a further report from the EMA highlighted the particular importance, for both ethical and methodological reasons, of cross-border cooperation between regulators and clinical research requirement compatibility for paediatric medicines [6]. The off-label prescription which is justified by clinical practice is a frequent practice when therapeutic alternatives are not available for the paediatric population group. This evidence-based off-label prescription has been recently described by a novel joint policy statement in Europe, with the purpose to offer guidance for health care professionals on when and how to prescribe off-label medicines to children and to provide recommendations for future European policy [7].

In Spain, in a two-year (2015 and 2016) cross-sectional study carried out for the medicines prescribed in primary health care paediatric patients, off-label use of drugs remained a public health concern especially for neonates and infants, who received the greatest proportion of off-label drugs [8]. To our knowledge, no study has been undertaken to determine the influence that the Paediatric Regulation adoption could have had on the prevalence of off-label use of medicines at the primary health care level of a member state. Thus, the main aims of the present study were to conduct a pre-post comparison on the annual off-label prescription rates in the under-18 population in Spain and assess the potential influence of the latest Paediatric Regulation (1901/2006/EC) adoption. In addition, a brief description of the legal framework in Spain for each study period was carried out, in relation to prescription in medicines on primary health care.

## 2. Materials and Methods

### 2.1. Study Design and Setting

An observational study of off-label prescription in population under 18 years of age at primary health care in Spain. Four annual periods were compared: one prior to the publication of the Paediatric Regulation (1901/2006/EC) (October 2004–September 2005) and three corresponding to the latest available periods following the adoption of the Regulation (October 2017–September 2018, October 2018–September 2019 and October 2019–September 2020).

General practitioners (GPs), paediatricians, auxiliary nurses, social workers, dentists and administrative staff are the backbone of the primary health care workforce in Spain [9]. According to the latest data available (2018), the Spanish National Health System has 143,995 health professionals, of which 35,486 (28,980 GPs and 6506 paediatricians) work in primary health care, which represents a rate of 0.8 per 1000 inhabitants [10].

### 2.2. Population and Eligibility Criteria

The target population is prescriptions for children and adolescents between the ages of 0 and 17. According to the official national data, the population between 0 and 17 years old in Spain increased from 7.6 million in January 2005 to 8.3 million in January 2020. This population group accounted for 18% of total population in the country for the four study periods [11].

Inclusion criteria: All prescriptions issued by primary care paediatricians, targeting all inhabitants aged 0–17 years old treated as outpatients in primary health care.

Exclusion criteria: All prescriptions issued in hospitals (inpatients) and by primary health care professionals other than paediatricians, such as GPs, nurses and other health professionals.

### 2.3. Data Sources

Study data collection included the prescription records from a private database (complete national scope) sorted by age group and drug (brand name, active ingredient, strength, package and dosage form) [12]. The primary source to determine off-label status was the General Pharmaceutical Council of Spain database (BotPlus 2.0) [13], which included an updated list of medicines which are not authorized for use in those under the age of 18. This database, which is available online, allows one to design and download a file with the medicines selected under this condition. The Official State Gazette of Spain (BOE) was also consulted online as a regulation source.

### 2.4. Variables

Variables collected were quantitative: demographic (age group) and resource use (number and percentage of prescriptions: total, off-label, by brand name, by active ingredient and by age group) and also qualitative: active ingredients and pharmacological subgroups (name and Anatomical Therapeutic Chemical Classification System (ATC) code) [14].

A prescription is the official form used to prescribe a medicine. According to the Spanish regulation, a single prescription could be used to prescribe several packages of the same medicine in specific situations (i.e., insulins and some systemic antibiotics).

Four age groups were defined: <1 year old (full-term neonates and infants), 1–<2 years old (toddlers), 2–11 years old (children) and 12–<18 years old (adolescents) [15].

In the present study, off-label drugs were defined as those that were not authorized for an age group according to the information in the BotPlus 2.0 database. Otherwise, they were on-label. For brand name prescriptions, the match was applied according to brand name in BotPlus 2.0, and in the case of active ingredient prescriptions, the match was applied according to active ingredient in BotPlus 2.0; in the latter, if the active ingredient had been included either in a branded product or in a generic one, it was considered as a possible off-label prescription and then counted.

### 2.5. Data Analysis

A descriptive analysis was carried out. Total number of prescriptions were studied according to labelling and drug characterization. Quantitative variables were published as totals and percentages. Percentages were also calculated for the characterization of the most prescribed off-label drugs.

Off-label assessment was carried out by two researchers independently; conflicts were resolved through discussion and consensus or consultation with a third reviewer. According to the BotPlus 2.0 database and the information included for off-label use in the under-18 population, some inconsistencies were detected, which led to some assumptions that were made during data analysis to address them (Table 1).

A statistical analysis of the data was also performed. The Odds Ratio (OR) of the number of off-label prescriptions in relation to the total number of prescriptions was calculated with 95% Confidence Intervals (CI). Also, the number of off-label prescriptions by brand name or active ingredient in relation to the total number of off-label prescriptions was considered. The year prior to the legislative change (October 2004–September 2005) was compared to the aggregate data of the three study periods after the legislative change (from October 2017 to September 2020).

## 3. Results

### 3.1. Total Prescriptions and Off-Label Rates in the Under-18 Population: Global Results

Overall, more than 150 million prescriptions were assessed for off-label use in the under-18 population during the four study periods (Table 2). Prescriptions by brand name were the most commonly issued by paediatricians, accounting for more than 70% of total prescriptions in all study periods.

The number of off-label prescriptions issued by paediatricians in the target population were estimated to be higher than 2 million per year (Table 2). Prior to the adoption of the Paediatric Regulation (1901/2006/EC), the off-label prescription rate was estimated at 7% of total prescriptions (3% of prescriptions by brand name and 4% of prescriptions by active ingredient). Although the 7% off-label rate remained almost unchanged during all study periods, off-label brand name prescriptions increased to 6% and those by active ingredient decreased to 2% in the latest study period (October 2019–September 2020).

After a pre-post comparison, the increase in off-label prescriptions was mild (OR: 1.045) but showed statistical significance (95% CI: 1.043–1.046; *p* < 0.05) after the Paediatric Regulation (1901/2006/EC) [1] adoption.

Regarding off-label prescriptions by brand name or active ingredient, a significant increase was observed in brand name prescriptions (OR: 2.431; 95% CI: 2.426–2.436; *p* < 0.05) and a subsequent decrease in off-label prescriptions by active ingredient (OR: 0.171; 95% CI: 0.171–0.172; *p* < 0.05).

An overview of the legal framework in Spain for each study period is included in Table 3. Basically, it describes the general rules regarding the prescription of medicines in the outpatient setting. Prioritization of prescriptions by active ingredient in Spain was introduced starting with Royal Decree-Law 16/2012 [16], which contributed to the former Royal Legislative Decree 1/2015, the one that was in force during the final three study periods (after the Paediatric Regulation (1901/2006/EC) publication).

The most relevant therapeutic groups and the active ingredients that accounted for the highest number of off-label prescriptions (<55% and >70%, respectively) were analyzed (Table 4 and Table 5).

Focusing specially on prescriptions by brand name, very few examples were identified of new drugs authorized from 2016 on, i.e., ten years after the publication of the Paediatric Regulation (1901/2006/EC).

During the three last study periods, an average of 356 different brand names were prescribed yearly in the under-18 population. Of these, only five were authorized after the year 2016 and were mostly indicated for dermatological conditions (*n* = 3). However, only one of the active ingredients was a real innovation (i.e., ozenoxacin, a topical antibiotic for the treatment of impetigo), as the others were additional new brands for existing medicines (i.e., ciclopirox, mometasone, prednisolone and combinations of calcipotriol).

### 3.2. Total Prescriptions and Off-Label Rates in the Under-18 Population: Results by Age Group

The age group that accounted for more than 60% of total prescriptions in all study periods was children between 2 and 11 years old, followed by neonates and infants (<1 year old), who accounted for 14–16% of all prescriptions in the under-18 population (Table 6).

The only age group that showed an increase in the percentage of off-label prescriptions after the Paediatric Regulation (1901/2006/EC) adoption was the under-1 population, whose rate increased 325%, from 4% (October 2004–September 2005) to 17% (October 2019–September 2020) over total prescriptions in the under-1 population. Likewise, the off-label prescription rate was initially estimated at 10% of total prescriptions in the population under 18, but it has been increasing up to 34% in the latest period (a 240% increase) (Table 6).

According to the statistical analysis, two opposite effects were observed. On the one hand, there was a significant increase in the number of off-label prescriptions in the under-1 population (OR: 4.270; 95% CI: 4.253–4.287; *p* < 0.05) and a mild increase in the group of children between 1 and <2 years old (OR: 1.045; 95% CI: 1.039–1.051; *p* < 0.05). On the other hand, a downward trend was observed in the population aged 2 to 11 years (OR: 0.630; 95% CI: 0.629–0.631; *p* < 0.05) and 12 to <18 years (OR: 0.752; 95% CI: 0.750–0.755; *p* < 0.05).

## 4. Discussion

### 4.1. Total Prescriptions and Off-Label Rates in the Under-18 Population: Global Results

Prescription of medicines by brand name was the most common practice among paediatricians (i.e., 85%, 71%, 73% and 73% respectively, depending on the study period). Before July 2006, the local regulation in force had not started protecting generic drugs and promoting prescription by active ingredient, which could explain the higher rate of prescription by brand name during this period (85%). This trend slightly decreased in the following periods, which could be influenced by the new regulation or could be coincidental.

The off-label prescription rate variation was mild in pre-post comparison, as it was between 6% and 8%, but significant (*p* < 0.05), which indicated that the Paediatric Regulation (1901/2006/EC) [1] adoption did not have any impact in this regard for primary care paediatricians. Conversely to total prescriptions, off-label prescriptions by brand name increased throughout the study periods (from 3% to 6% of total prescriptions in the under-18 population). Consequently, the risk of off-label prescription did not appear to be influenced by either the protectionist regulation endorsing prescription by active ingredient in Spain [18] or by the adoption of the Paediatric Regulation (1901/2006/EC) [1] in the EU.

Regarding the extent of off-label use in EU, the systematic literature review undertaken by Weda et al. (2017) [5] showed a wide range of off-label rates (2–100%) in the outpatient setting for children (≤18 years) after analyzing 40 studies. In addition, wide variation between countries was observed and no clear patterns were identified. Under this scenario, the results from our study would be positive, being on the lower end of the range (<10%). The only study from Spain that was included in this publication, Ruiz-Antorán et al. (2013) [19], pointed to a prevalence of off-label use of 33.2%.

A recent systematic literature review of off-label medication use in the population <18 years identified 31 studies (in a mix of inpatient and outpatient settings), with off-label prescription rates from 3.2% to 95% [20]. However, most of the studies that aimed to determine the prevalence of off-label use in Spain and drug characterization in the population under 18 were mainly focused on at hospital settings [21,22] and could not be compared to those presented here.

According to the results of the 2012–2013 paediatric national survey of off-label drug use in children in Spain (OL-PED) [23], a significant percentage of paediatricians went on prescribing medications in their daily clinical practice without knowing whether the drug was indicated in children or for certain age groups. 673 responses were received. 75% of paediatricians knew the meaning of off-label use, 61% of them prescribed off-label medicines and just a small percentage (22% of them) wrote it down in the patient’s medical record as recommended by the Spanish legislation [24]. Although this survey was carried out after the adoption of the Paediatric Regulation (1901/2006/EC) [1], the authors indicated a need to adopt measures to raise awareness of the use of off-label (e.g., spontaneous notifications). To our knowledge, no further follow-up has been published on this survey.

In the EU study in 2017, the most common therapeutic areas in which off-label use in children was identified were cardiovascular diseases (e.g., antihypertensives), infectious diseases (e.g., antibacterial drugs), central nervous system (e.g., analgesic agents), respiratory system (e.g., asthma medicines) and alimentary tract and metabolism drugs (e.g., reflux medicines). In the case of the present study, those pharmacological subgroups with the highest number of off-label prescriptions were focused on the alimentary tract and metabolism (e.g., peptic ulcer and gastro-esophageal reflux disease; intestinal anti-inflammatory agents; vitamin A and D), analgesics (e.g., other analgesics and antipyretics), the respiratory system (e.g., decongestants and other nasal preparations for topical use; antihistamines for systemic use; other systemic drugs for obstructive airway diseases), the musculo-skeletal system (e.g., anti-inflammatory and antirheumatic products, non-steroids), and the sensory organs (e.g., corticosteroids and anti-infectives in combination).

More than 70% of off-label prescriptions for outpatients in Spain consisted of active ingredients that had been in the market for decades (e.g., ibuprofen, paracetamol, ketoconazole, domperidone, colecalciferol or budesonide), which excluded them from being in the primary scope of the latest Paediatric Regulation (1901/2006/EC) in Europe. Consequently, the changes introduced by this Regulation in terms of new product development and authorization could have had little or no impact in off-label prescription rates throughout the study period. This scenario could also reveal a pattern: Spanish paediatricians in the primary health care setting are not early adopters of innovation. This pattern could be justified by multiple reasons (i.e., high price of innovations, uncertainty over the lack of long-term data about effectiveness and safety), although they are out of the scope of the present study. What is clear is that regardless of being off-label medicines according to the SmPC, paediatricians in the primary health care setting in Spain seem to rely strongly on the existing post-authorization clinical evidence to prescribe in the population under the age of 18. Other source of additional information for prescription could be the American label authorized by the Food and Drug Administration (FDA), since sometimes the same medicines have different indications depending on the geographical location and/or medicines agency involved, thereby leaving the door open for off-label use.

### 4.2. Total Prescriptions and Off-Label Rates in the Under-18 Population: Results by Age Group

Patients under 1 year of age are the most vulnerable segment of the population with regards to adverse drug events in case of off-label drug use [5,23]. Following this statement, it should be noted that the results of the present study pointed out that off-label prescriptions increased significantly in the population under 2 after the adoption of the Paediatric Regulation (1901/2006/EC). Additionally, the recent systematic literature review from Almeida et al. (2020) [25] found that the highest proportion of unlicensed prescription in primary health care setting was in children under 2 years of age.

This increase was dramatically higher in those under 1 year of age, which raises a red flag regarding the management of this population group, increasing its risk and vulnerability. Despite the fact that the total number of prescriptions issued by paediatricians for this age group decreased throughout the study periods, the off-label prescription rate greatly increased. Considering the total prescriptions in this age group, the off-label rate increased fourfold throughout the first and last study periods, specifically from 4% to 17%. The difficulty of dose determination and new drug development programs for its use in neonates is of utmost importance. Some diseases are unique to neonates, but others could be similar (or not) to other population groups, which adds complexity to this end. In addition, several factors should be considered for specific drug dosing approaches, as maturational pharmacokinetics and body size, which play a unique role in this subpopulation group [26].

Although off-label use risks compromising patients’ safety, many experiences of use in new indications and age groups are supported by the literature and by real-world evidence. Formally, the Summary of Product Characteristics (SmPC) is the legal document approved as part of the marketing authorization of each medicine which is the primary source to determine off-label prescription. It is based on pivotal randomized clinical trials and its modification requires additional and time-consuming procedures of evaluation with the regulatory bodies, based on new highly controlled randomized clinical trials.

However, there are alternative sources of robust information, for instance, the British National Formulary, Pediamécum (a Spanish documentary database of active substances used in Paediatrics) or DailyMed in the United States. These databases collect and validate updated clinical practice, such as the off-label use of medicines. Most health professionals rely on the information gathered in these databases and use it in their current clinical practice. This situation exposes the gap that exists between real-world evidence or real clinical practice and the regulatory basis of medicines use, which may lead to the observed impairment between the results of this study and the current legal framework.

Finally, the fact that during the three last study periods, only one of the active ingredients was a real innovation (ozenoxacin) reflects that very few new medications are finally authorized and marketed to be prescribed by primary care paediatricians. Drugs of most interest in this study (e.g., ibuprofen, paracetamol, colecalciferol, budesonide, etc.), which are included in Table 4, have been in the market for decades, which excludes them of being in the primary scope of the latest Paediatric Regulation (1901/2006/EC) in Europe, focused on new product development and authorization. In this context, maybe research and development of medicines adapted to paediatric needs should not only comprise innovations but also mature products in future initiatives. On the other side, the access of innovations could also be limited for certain innovations in primary health care in Spain (e.g., highest prices than the standard of care, uncertainty in effectiveness), which would slow down its introduction in clinical practice [5].

### 4.3. Limitations of the Study

This study has some limitations that could lead to an estimation bias of off-label use (low incidence). First, this study was focused on the off-label use of authorized medicines and did not include medical devices, compounded formulas, cosmetics, food supplements and medicinal plants. Second, only prescriptions issued by paediatricians in primary health care were considered; prescriptions from another medical specialties or those derived from hospitalizations were excluded. Third, non-prescription medicines were not taken into account (i.e., self-medication). Fourth, as the data analyzed was not linked to medical records, it was not possible to assess the indication, the dose or if the off-label use was justified by the paediatricians. Fifth, results show potential off-label use from prescriptions, as dispensation data was not provided by the database used. Sixth, the information about off-label drugs in the used database does not contain a breakdown by ages, meaning that it considers the under-18 population as a whole; subsequent analysis by SmPC information will be needed as a further step to refine results by age. Regretfully it is not possible to do this automatically, so it is not feasible to analyze case by case for such a huge number of prescriptions analyzed (>150 million). Finally, our primary source of information lacks accurate information about total prescriptions in Spain for the period prior to 2005, which prevented increasing the sample of pre-follow-up periods and upgrading the statistical analysis. Nevertheless, the highest number of prescriptions assessed for off-label use, accounting for the total paediatric population (0–17 years) in Spain for four years, adds value to this project and the results disclosed previously.

## 5. Conclusions

This study did not find a decrease of the annual off-label prescription rate in the primary health care setting of the Spanish paediatric population, even though the latest Paediatric Regulation (1901/2006/EC) publication, and more than 10 years after its adoption. The most likely explanation for this gap is the scarce prescription of innovations in primary health care setting, which could benefit from the initiatives introduced by this regulation. Conversely, most prescriptions (>70%) issued for the paediatric outpatients included in the study consisted of active ingredients that had been in the market for decades and consequently were not under the scope of this regulation in terms of paediatric development for a safer use of medicines in the population under 18.

It should be noted that neonates and infants (<1 year old) accounted for 14–16% of all prescriptions in the under-18 population and together with those between 1 and <2 years old were the only age groups that also showed a relevant increase in the off-label prescription rate. The vulnerability of these age groups and the global findings of the present project can help raise awareness and advocate for the development of safer medicines for children. Future initiatives and legislative tools should encourage research and development of paediatric medicines focused on the areas of highest unmet need, also in the primary healthcare setting, promoting the international cooperation of regulatory bodies across the globe and speeding up the time to market.

## Figures and Tables

**Table 1 pharmaceutics-13-00588-t001:** Definition of inconsistencies that were identified in the BotPlus 2.0 database according to off-label indication in the under-18 population.

Inconsistency Description	Examples	Assumption/Decision
Medicines with the same active ingredient, similar formulation and administration route, where low doses were classified as off-label but not high doses.	Pantoprazole and Omeprazole 20 mg (tablets or capsules) were defined as off-label in the under-18 population but not 40 mg.	Pantoprazole and Omeprazole 20 mg (tablets or capsules) were considered off-label in the under-18 population.
	Paracetamol 500 mg effervescent tablets and 1 g tablets (effervescent or not) were defined as off-label in the under-18 population, but not 500 mg tablets or capsules.	Paracetamol 500 mg tablets and capsules were considered off-label in the under-18 population.
Topical medicines with the same active ingredient, same dose or concentration and administration route, where some formulations were classified as off-label, but others were not depending on the formulation.	Similar formulations: Ketoconazole cream was defined as off-label in the under-18 population but not the gel, which is also a semi-solid. Both were formulated with the same concentration (2%; 20 mg/g).	Both Ketoconazole cream and Ketoconazole gel were considered off-label in the under-18 population.
Topical medicines with the same active ingredient, different dose or concentration and same administration route, where some formulations were classified as off-label, but others were not depending on the formulation.	Different formulations: *Centella asiatica herba* in combination with antibiotics formulated as powder (solid) was defined as off-label in the under-18 population but not the ointment (semi-solid). Both were formulated with a different concentration (powder, 1%; ointment, 2%).	Only *Centella asiatica herba* powder was considered off-label in the under-18 population.
Topical medicines with the same active ingredient, same dose or concentration and administration route, where some formulations were classified as off-label, but others were not depending on whether they were brands or generics.	In the case of Ciclopirox, the variability in off-label designation depended on the brand name, ranging from nail lacquer to shampoo, topical solution and cream.	For Ciclopirox brand names, the off-label designation was assessed following BotPlus 2.0 criteria brand by brand in the under-18 population.
		In the case of Ciclopirox generics, all formulations have been assessed as potentially off-label in the under-18 population.
Medicines with many generics in the market, where some were classified as off-label, but others were not.	Many generics of Paracetamol 1 g and 500 mg tablets.	Due to the high number of cases identified under this condition, all Paracetamol 1 g and 500 mg prescriptions by active ingredient were assessed as potentially off-label in the under-18 population.

**Table 2 pharmaceutics-13-00588-t002:** Number of prescriptions and off-label rates per study period in the under-18 population.

Period (MAT)	Prescriptions (Total), *n*	Prescriptions by Brand Name, *n* (%) *	Prescriptions by Active Ingredient, *n* (%) *	Off-Label Prescriptions (Total), *n* (%) *	Off-Label Prescriptions by Brand Name, *n* (%) *^,†^	Off-Label Prescriptions by Active Ingredient, *n* (%) *^,†^
October 2004–September 2005	44,348,661	37,916,646 (85%)	6,432,015 (15%)	2,924,240 (7%)	1,171,325 (3%; 3%)	1,752,915 (4%; 27%)
October 2017–September 2018	37,698,501	26,838,326 (71%)	10,860,175 (29%)	2,098,955 (6%)	1,372,365 (4%; 5%)	726,590 (2%; 7%)
October 2018–September 2019	37,135,180	26,974,055 (73%)	10,161,125 (27%)	2,686,781 (7%)	2,154,107 (6%; 8%)	532,674 (1%; 5%)
October 2019–September 2020	32,385,213	23,546,363 (73%)	8,838,850 (27%)	2,577,342 (8%)	2,037,325 (6%; 9%)	540,017 (2%; 6%)

* Percentage calculated over total prescriptions in the under-18 population; ^†^ Percentage calculated over prescriptions by brand name or active ingredient in the under-18 population. Abbreviations: MAT, moving annual total (12-month period).

**Table 3 pharmaceutics-13-00588-t003:** Local regulation and general rules regarding the prescription of medicinal products during each period of the project.

Period (MAT)	Regulation in Force	Comments
October 2004–September 2005	Law 25/1990 [17]	Aim: The primary objective is to contribute to the provision of safe, effective and quality medicines, properly identified and with appropriate information. Prescription of medicines: No specific references to the prioritization of generics over branded medicines are included.
October 2017–September 2018	Royal Legislative Decree 1/2015 [18]	Aim: To guarantee the quality of all benefits being provided by the Spanish National Health System, ensuring better protections for the rational use of medicines and that access to medicines is done through a more effective system with tighter safety controls. Prescription of medicines: The general rule is to prescribe by active ingredient whereby the least expensive medicinal product within its homogeneous group * will be supplied and, if the prices are the same, the appropriate generic or biosimilar medicinal product will be supplied. - In acute conditions and the first prescription for chronic conditions, prescriptions will, generally, be issued by active ingredient. - Brand names are allowed to be prescribed, subject to the principle of greater efficiency for the system or where the medicinal products concerned are regarded as being non-substitutable. - If the price of a medicinal product prescribed by brand name is higher than the lowest priced product in its homogeneous group, the pharmacist must substitute the prescribed product with the lower priced product and, if the prices are the same, must supply the appropriate generic or biosimilar medicinal product.
October 2018–September 2019		
October 2019–September 2020		

* Spanish regulation groups outpatient medicines which are reimbursed by the National Health Service into “homogeneous groups”. Each homogeneous group is formed by pharmaceuticals with the same active ingredient (regardless of being branded or generic), dose, content, pharmaceutical form and route of administration, which can be interchangeable. The lowest price of each homogeneous group is the maximum price paid by the National Health Service. The lowest price is updated each month. Abbreviations: MAT, moving annual total (12-month period).

**Table 4 pharmaceutics-13-00588-t004:** Number of off-label prescriptions for the top 10 most-prescribed active ingredients per study period in the under-18 population.

Period	Active Ingredient	Off-Label Prescriptions (Total), *n* (%) *	
October 2004–September 2005	Ibuprofen	1,092,261	(37%)
	Paracetamol	759,027	(26%)
	Silicones	130,981	(4%)
	Dexamethasone and anti-infectives	129,455	(4%)
	Budesonide	125,352	(4%)
	Methylprednisolone aceponate	119,932	(4%)
	Acetylcysteine	78,621	(3%)
	Benzydamine	49,359	(2%)
	Ketoconazole	38,289	(1%)
	Domperidone	29,477	(1%)
October 2017– September 2018	Colecalciferol	503,925	(24%)
	Paracetamol	241,082	(11%)
	Budesonide	142,668	(7%)
	Omeprazole	118,773	(6%)
	Ibuprofen	113,440	(5%)
	Ketoconazole	93,149	(4%)
	Acetylcysteine	90,283	(4%)
	Methylprednisolone aceponate	84,551	(4%)
	Dexamethasone and anti-infectives	52,953	(3%)
	Terbinafine	45,736	(2%)
October 2018–September 2019	Colecalciferol	623,089	(23%)
	Mepyramine theophyllinacetate	579,251	(22%)
	Paracetamol	224,193	(8%)
	Budesonide	116,142	(4%)
	Omeprazole	95,049	(4%)
	Dexamethasone and anti-infectives	72,970	(3%)
	Acetylcysteine	63,630	(2%)
	Methylprednisolone aceponate	61,324	(2%)
	Ketoconazole	60,069	(2%)
	Silicones	59,920	(2%)
October 2019–September 2020	Colecalciferol	658,774	(26%)
	Mepyramine theophyllinacetate	360,325	(14%)
	Dexchlorpheniramine	199,125	(8%)
	Paracetamol	197,504	(8%)
	Budesonide	105,940	(4%)
	Omeprazole	104,163	(4%)
	Methylprednisolone aceponate	78,827	(3%)
	Ciclopirox	62,718	(2%)
	Dexamethasone and anti-infectives	60,289	(2%)
	Silicones	59,611	(2%)

* Percentage calculated over total off-label prescriptions in the population under the age of 18.

**Table 5 pharmaceutics-13-00588-t005:** Number of off-label prescriptions in the top 5 most-prescribed pharmacological subgroups per study period in the under-18 population.

Period	Pharmacological Subgroups (ATC Code and Description)		Off-Label Prescriptions (Total), *n* (%) *	
October 2004–September 2005	M01A	Anti-inflammatory and antirheumatic products, non-steroids	1,121,520	(38%)
	N02B	Other analgesics and antipyretics	760,808	(26%)
	A03A	Drugs for functional gastrointestinal disorders	130,981	(4%)
	S03C	Corticosteroids and anti-infectives in combination	129,455	(4%)
	R01A	Decongestants and other nasal preparations for topical use	123,399	(4%)
October 2017–September 2018	A11C	Vitamin A and D, incl, combinations of the two	505,517	(24%)
	N02B	Other analgesics and antipyretics	242,700	(12%)
	A02B	Drugs for peptic ulcer and gastro-esophageal reflux disease	159,644	(8%)
	M01A	Anti-inflammatory and antirheumatic products, non-steroids	151,669	(7%)
	A07E	Intestinal anti-inflammatory agents	148,852	(7%)
October 2018–September 2019	A11C	Vitamin A and D, incl. combinations of the two	646,194	(24%)
	R03D	Other systemic drugs for obstructive airway diseases	579,251	(22%)
	N02B	Other analgesics and antipyretics	224,193	(8%)
	A02B	Drugs for peptic ulcer and gastro-esophageal reflux disease	127,022	(5%)
	R01A	Decongestants and other nasal preparations for topical use	119,031	(4%)
October 2019–September 2020	A11C	Vitamin A and D, incl. combinations of the two	674,105	(26%)
	R03D	Other systemic drugs for obstructive airway diseases	360,325	(14%)
	R06A	Antihistamines for systemic use	199,125	(8%)
	N02B	Other analgesics and antipyretics	197,504	(8%)
	A02B	Drugs for peptic ulcer and gastro-esophageal reflux disease	137,866	(5%)

* Percentage calculated over total off-label prescriptions in the population under the age of 18. Abbreviations: ATC, Anatomical Therapeutic Chemical Classification System.

**Table 6 pharmaceutics-13-00588-t006:** Number of prescriptions and off-label rates per study period in the under-18 population by age group.

Period (MAT)	<1 Year Old		1–<2 Years Old		2–11 Years Old		12–<18 Years Old	
	Prescriptions (Total), *n* (%) *	Off-Label Prescriptions, *n* (%) **^,§^	Prescriptions (Total), *n* (%) *	Off-Label Prescriptions, *n* (%) **^,§^	Prescriptions (Total), *n* (%) *	Off-Label Prescriptions, *n* (%) **^,§^	Prescriptions (Total), *n* (%) *	Off-Label Prescriptions, *n* (%) **^,§^
October 2004 –September 2005	7,315,428 (16%)	287,438 (4%; 10%)	4,476,495 (10%)	173,453 (4%; 6%)	29,011,169 (65%)	1,882,158 (6%; 64%)	3,545,569 (8%)	581,191 (16%; 20%)
October 2017–September 2018	5,126,880 (14%)	545,457 (11%; 26%)	3,086,318 (8%)	75,746 (2%; 4%)	24,640,700 (65%)	857,149 (3%; 41%)	4,844,603 (13%)	620,603 (13%; 30%)
October 2018–September 2019	5,148,416 (14%)	871,102 (17%; 32%)	2,982,650 (8%)	141,300 (5%; 5%)	24,126,595 (65%)	1,025,690 (4%; 38%)	4,877,519 (13%)	648,689 (13%; 24%)
October 2019–September 2020	5,193,250 (16%)	883,044 (17%; 34%)	2,552,232 (8%)	131,328 (5%; 5%)	20,128,907 (62%)	1,002,657 (5%; 39%)	4,510,824 (14%)	560,313 (12%; 22%)

* Percentage calculated over total prescriptions in the population under the age of 18; ** Percentage calculated over total prescriptions in the population group; ^§^ Percentage calculated over total off-label prescriptions in the population under the age of 18. Abbreviations: MAT, moving annual total (12-month period).

## Data Availability

The data presented in this study are available from the corresponding author upon request.

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
