# Peer review of "Effects of the Off-Label Drug Prescription in the Paediatric Population in Spain from the Adoption of the Latest European Regulation: A Pre-Post Study"

_pharmaceutics, 2021, doi:10.3390/pharmaceutics13040588_

Round 1

Reviewer 1 Report

Dr. Lizano-Díez reported a study on the effects of off-label drug prescription in the pediatric population in Spain. The study is well-designed. The data collection seems sufficient. The presentation of the data is clear. The finding is also interesting, both for the general public, pharmacists, and health regulators. There are two shortages that I will like to bring up. One is the background of the study. The authors described the aim of the study in the Introduction. It would be better to put this into a broader background, such as the situation in the EU, which will draw attention to a broader readership. The second is the Discussion. Many results were not discussed in the section, which is a pity. I would suggest the author consider these two points and expand the discussion with a deeper and broader perspective.
However, I could suggest the manuscript being accepted in the current form.

Reviewer 2 Report

I highly value the effort made by the authors to quantify the off label practices before and since the European regulation. The issue of off label use in pediatrics remains a relevant topic, not limited to Spain, but across Europe (as this paper refers to the European regulation).  I miss the recent European paper (joint policy statement) on off label and unlicensed use of drug in children in Europe to put the paper into a broader perspective (Schrier et al, Eur J Pediatr 2020). This covers also the fact that off label does not equal off knowledge use of drugs.

Based on the methods section as described, the ‘only’ indicator assessed in the current study is the prescription of a given drug in <18 year patients and in primary care, and does not consider the indication or the dose (if I understood this well), nor hospitalized cases. If so, the trends are still relevant, but these are limitations that should be further considered (and likely explains the overall ‘low incidence’ of off label use when reading the absolute data), so that a ‘best case scenario has been compared before/after. As dosing is relevant,  I suggest to cross refer to the recent paper in this journal on dose finding in neonates (van den Anker et al. on neonatal drug development. Pharmaceutics 2020).

Just to be sure on this, but is a european registration immediately converted to the national legal setting, or are there additional national hurdles (registration, reimbursement).

Reviewer 3 Report

This study was to conduct a pre-post comparision on the annual off-label prescription rates in the under 18 population in Spain and assess the potential influence of the Paediatric Regulation adoption. The results may help raise awareness and advocate for the development and authorisation of medicines for children in the primary health care setting. The study is a very interesting, but the study have major concerns as below:

  1. The authors should explain why the study explain the paediatric population and  european regulation.
  2. I am wondering the differences of FDA and EMA in your aim of study. Please explain it.
  3. Please check the style of references.
  4. Please show the figure of study flow.

Round 2

Reviewer 3 Report

Thank you for the response of my comments. 

The authors should check plagiarism and edit the revised manuscript.
